# Smoothing County-Level Sampling Variances to Improve Small Area Models' Outputs

Lu Chen [1,2,*], Luca Sartore [1,2], Habtamu Benecha [2], Valbona Bejleri [2] and Balgobin Nandram [2,3]

1 National Institute of Statistical Sciences, 1750 K Street NW Suite 1100, Washington, DC 20006, USA
2 United States Department of Agriculture, National Agricultural Statistics Service, 1400 Independence Avenue SW, Washington, DC 20250, USA
3 Worcester Polytechnic Institute, Stratton Hall 103, Worcester, MA 01609, USA
* Correspondence: lchen@niss.org

**Abstract:** The use of hierarchical Bayesian small area models, which take survey estimates along with auxiliary data as input to produce official statistics, has increased in recent years. Survey estimates for small domains are usually unreliable due to small sample sizes, and the corresponding sampling variances can also be imprecise and unreliable. This affects the performance of the model (i.e., the model will not produce an estimate or will produce a low-quality modeled estimate), which results in a reduced number of official statistics published by a government agency. To mitigate the unreliable sampling variances, these survey-estimated variances are typically modeled against the direct estimates wherever a relationship between the two is present. However, this is not always the case. This paper explores different alternatives to mitigate the unreliable (beyond some threshold) sampling variances. A Bayesian approach under the area-level model set-up and a distribution-free technique based on bootstrap sampling are proposed to update the survey data. An application to the county-level corn yield data from the County Agricultural Production Survey of the United States Department of Agriculture's (USDA's) National Agricultural Statistics Service (NASS) is used to illustrate the proposed approaches. The final county-level model-based estimates for small area domains, produced based on updated survey data from each method, are compared with county-level model-based estimates produced based on the original survey data and the official statistics published in 2016.

**Keywords:** agricultural survey; Bayesian; bootstrap; small area estimation; unreliable variances





## 1. Introduction

Small area estimation methods include a wide range of modeling techniques generally devised to improve estimates for domains where direct estimates are not reliable due to small sample sizes. Area-level models are typically used in small area estimation under the assumption that the survey data and the auxiliary information at the area level are available. For continuous responses, the first and most commonly used area-level model is the Fay–Herriot (FH) model [1]. A key assumption of the FH model is that the survey-estimated variances are fixed and known. However, the sampling variances for small domains can vary and be unreliable (beyond some threshold) due to very small sample sizes. This affects the performance of the model (i.e., the model will not produce an estimate or will produce a low-quality modeled estimate), which results in a reduced number of official statistics published by a government agency. This paper proposes and explores two alternatives to mitigate the unreliable survey-estimated variances. A Bayesian approach under the area-level model set-up and a distribution-free technique using bootstrap sampling to update the estimates from the survey are presented. By smoothing the sampling variances (further used as hierarchical Bayesian (HB) small area model inputs), the proposed methods improve the overall performance of the HB small area models. Both algorithms are general in nature

and can be applied to surveys suffering from small area issues, such as unavailable or unreliable survey summaries due to the volatility of sampling variances associated with the survey design. For ease of understanding, we illustrate the two methods with data from the United States Department of Agriculture's (USDA's) National Agricultural Statistics Service (NASS) crops county estimates program.

The USDA's NASS conducts the County Agricultural Production Survey (CAPS) to obtain end-of-year estimates of the total planted acreage (P), total harvested acreage (H), total production (G) and yield (Y) for dozens of small grains and row crops at the county level. Yield is defined as the ratio of total production to the harvested acreage. Starting in 2020, several HB subarea-level (small area) models have been implemented as extensions of the FH model to improve the precision of the estimates at the county level (see [2–6]). The sampling variances of the yield estimates are produced using a second-order Taylor series approximation and, due to various reasons (e.g., sparseness in data), could result in zero, very small or very large estimated variances for several counties. This contradicts one of the model's assumptions, which is "the sampling variances are fixed and known". In the 2016 CAPS, the sampling variances of corn yield were very small for approximately 10% of the counties [7]. The volatility of the variances estimated from surveys directly affects the estimates obtained from the HB models for the corresponding counties. It also indirectly affects the estimates produced through HB models for other neighboring counties. This ultimately results in a reduced number of published modeled county estimates for the US by the NASS.

Often, direct point estimates from surveys are not reliable for areas with small sample sizes, and the corresponding sampling variances can also be imprecise and inaccurate [8]. The choice and accuracy of the existing techniques for small area estimation are dictated by the quality of the available data in terms of both survey responses and covariates (see [9]). The literature suggests different approaches under both frequentist and Bayesian frameworks to mitigate unreliable survey-estimated variances. One common approach, known as the generalized variance function (GVF), includes modeling survey-estimated variances against some function of the direct estimator and other covariates [10] to smooth the noise and make the variances stable. Researchers have proposed co-modeling of direct estimates from the surveys and their variances within a model. For example, Maiti et al. [11] used a frequentist framework, and Sugasawa et al. [12] considered a Bayesian approach, which was further extended by Gershunskaya and Savitsky [13] to include nonparametric probabilistic clustering. A hierarchical log-normal model was developed by Erciulescu et al. [14] to mitigate unreliable sampling variances that were strongly related to direct estimates. Then, the updated sampling variances were further used as input in a subarea HB model for producing the corn harvested acreage. The log-normal model is also used to smooth the survey-estimated variances of production totals since its assumptions hold for the production totals too. However, this model's assumptions do not hold for the yield. In addition, exploratory data analyses do not indicate any relationship between the direct estimates of the yield and their sampling variances that could easily be modeled. It is not always easy to model the relationship between survey summaries (direct estimates and sampling variances) using classical approaches due to the confounding factors involved. One can use a Taylor series approximation under a subarea-level model, as shown in [7,14], to update the variances in the yield.

The question of interest is how to mitigate the unreliable sampling variances in scenarios where the relationship between the sampling variances and direct estimates cannot be modeled using classical approaches. Using a case study from the 2016 CAPS, Bejleri et al. [7] explored alternative approaches other than modeling, such as Taylor series approximation and a data-driven technique using bootstrap sampling to mitigate survey variances in the yield produced as zeros or below a threshold (1 bushel per acre). Measures of uncertainty were produced from each approach for all US counties in the sample with valid (positive) direct estimates of the yield. Even though the number of final modeled county estimates increased, applying these alternative approaches to CAPS did not adjust

for all extreme outliers of the sampling variances (i.e., extreme values in the upper tail of the distribution were not updated by this method).

In this paper, a strategy that considers smoothing all extreme outliers of the sampling variances to improve the performance of the HB small area models (which take survey summaries and other auxiliary data as input) is introduced wherever the relationship between the survey-estimated variances and direct estimates cannot be modeled using classical approaches. Two alternative methods for mitigating unreliable (beyond some threshold) survey-estimated variances of the yield at the county level are explored. Bayesian modeling using a non-informative prior (hereafter called the Bayesian method) and an empirical approach based on bootstrap sampling (hereafter called the bootstrap method) to update the sampling variances for small domains are presented. The bootstrap sampling is used to construct the empirical distribution of survey-estimated variances. The two proposed approaches described in Section 3 smooth all unreliable sampling variances into reliable inputs for the HB models. Steps describing the methodology discussed in this paper for mitigating unreliable survey-estimated variances are presented in Figure 1. For ease of understanding and illustration, the two methods are applied to data from USDA's NASS crops county estimates program (Section 4). The results from the case study in Section 4 show substantial improvement in the modeled estimates produced based on the updated sampling variances and an increase of approximately 10% in the number of final model-based county estimates produced.

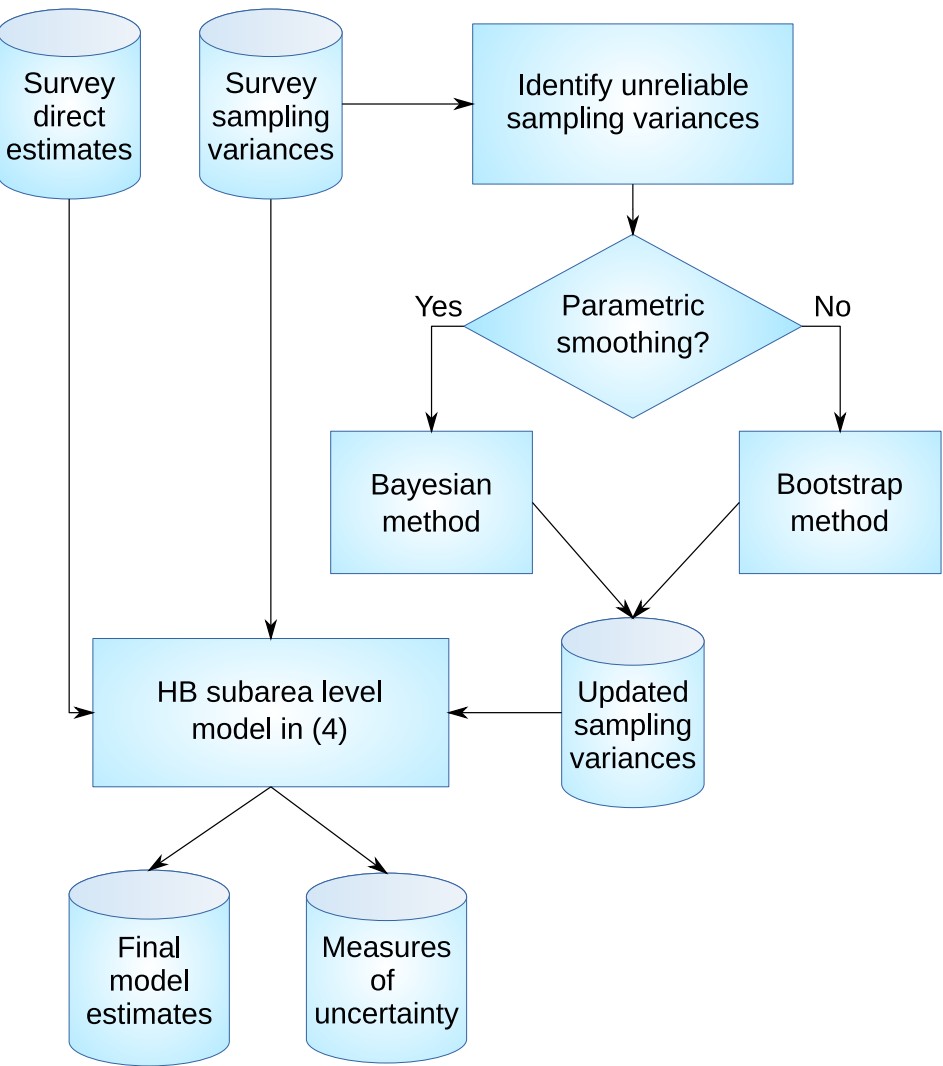

**Figure 1.** Flowchart describing the estimation process.

This paper is structured as follows. The motivation for this work and some background are presented in Section 2. The CAPS and the relationship between the survey's direct estimates and their survey-estimated variances are briefly discussed. The methodology for generating (updated) survey variances in the yield for small domains is presented in Section 3. The proposed Bayesian and bootstrap methods for updating the survey variances are presented in detail. The county estimates produced by HB models, using as input the direct estimates and the improved sampling variances derived from each approach, are compared with the published official county statistics, and the results are discussed in Section 4. The concluding remarks are presented in Section 5.

## 2. Motivation and Background

### 2.1. County Agricultural Production Survey

NASS has been producing official county-level crop inventories since 1917, but in 2011, the NASS instituted major changes to improve the quality of the program. It implemented the large-scale probability survey CAPS to provide county-level official estimates for 13 principal small grains and row crops in many states annually. County-level estimates provide public benchmarks for farmers, ranchers and the industry, and they also serve as key indicators of a number of federal and state agencies for farm policy formation, program implementation and management. The CAPS list frame samples are selected using a multivariate probability proportional to size (MPPS) sampling scheme in which the measure of size is determined by more than one item. Sampling variances for the totals (P, H and G) are estimated using a delete-a-group Jackknife (DAGJ), and the sampling variances for the ratios (Y) are estimated using second-order Taylor series approximation for the ratio (see [15]). Starting in 2020, NASS has used HB small area models, which use survey direct estimates, sampling variances and auxiliary information as inputs to produce the model-based estimates as the key indicators for the official statistics at the county level.

In this paper, we use the corn yield estimates from the 2016 CAPS as the case study. The 2016 CAPS sample consists of 37 states comprising 2881 counties for corn. From these counties, 2467 had positive planted acreages, 2361 counties had positive harvested acreages, and 2329 counties had positive yields or production for corn. Exploratory data analysis of the CAPS responses for corn yields in 2016 revealed that among the counties with positive direct estimates, there were 241 counties with sampling variances ($\hat{\sigma}_i^2$) equal to zero or that fell 1.5 $IQR(\hat{\sigma}_i^2)$ below the first quartile and 107 counties with sampling variances that were relatively large (i.e., that fell 1.5 $IQR(\hat{\sigma}_i^2)$ above the third quartile), where $IQR$ stands for the interquartile range. Let $L = \hat{Q}_{0.25} - 1.5\ IQR(\hat{\sigma}_i^2)$ and $U = \hat{Q}_{0.75} + 1.5\ IQR(\hat{\sigma}_i^2)$. The use of these types of boundaries ($L$ and $U$) has proven to be beneficial for outlier detection (see [16,17]). We consider these boundaries throughout the paper to identify "anomalous" counties. There were 348 counties with positive survey estimates of yields that fell outside of the $L$ and $U$ boundaries in total, comprising more than 10% of all counties.

This paper addresses the question of interest (i.e., how to mitigate the unreliable sampling variances) by adjusting for all extreme outliers of the survey-estimated variances. Two approaches to mitigating the unreliable sampling variances that fall outside of the boundaries ($L$ and $U$) are introduced.

### 2.2. Exploring the Relationships between Direct Estimates and Their Variances

Exploratory data analysis of the 2016 CAPS summaries for corn from the sampled US counties revealed a strong relationship (on a log scale) between the direct estimates and their variances for P, H and G (see Figure 2). The relationship between the survey's direct estimates and their variances is modeled whenever a "good" relationship between the two is present. A hierarchical log-normal model for the sampling variances of the survey estimates was developed by Erciulescu et al. [14] to mitigate the zero variances for planted acres, harvested acres and the production of corn. The coefficients were estimated using the subset of sampled data, with survey estimates available (non-zero) for both quantities. However, the log-normal model assumptions did not hold for the yield. The plot of the

survey-estimated variances against the direct estimates of the yield for the sampled US counties (in Figure 2) does not suggest any relationship that could be modeled using the classical approaches suggested in the literature. We concentrate on sampling variances that do not satisfy the assumptions of the log-normal model and use the 2016 CAPS corn yield as a case study to illustrate the approaches proposed in this paper.

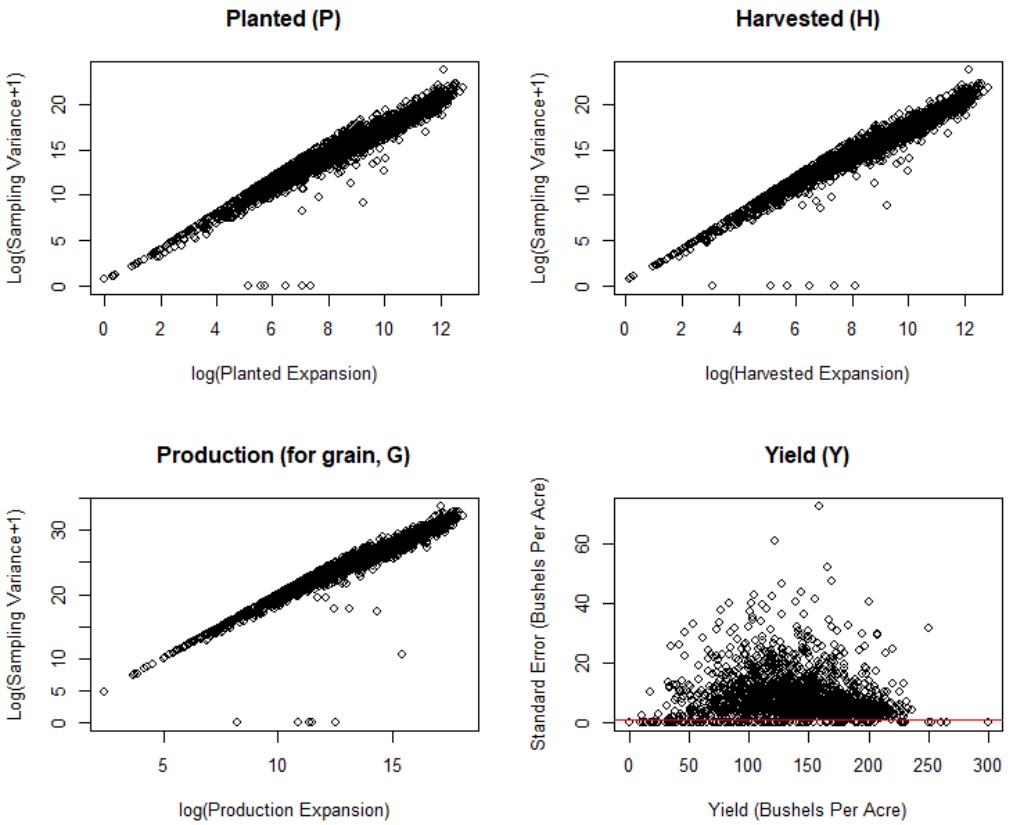

**Figure 2.** Direct estimates of total planted acres, harvested acres, production and yield for sampled US counties in log scale.

## 3. Generating Survey Variances of the Yields for Anomalous Counties

In this section, we propose two alternative approaches for smoothing the survey-estimated variances of the yield at the county level (i.e., generate new ones) for the counties with survey variances that fall outside the threshold bounds (hereafter called "anomalous counties"). A Bayesian method using a non-informative prior and a bootstrap method to update the survey variances for small domains are presented in the next two subsections. The first approach, a Bayesian method, uses the HB model setting. The second, a bootstrap method, is a data-driven approach that utilizes bootstrap sampling to construct the empirical distribution of variances estimated from the survey. This empirical distribution is further used to generate updated variances for the anomalous counties. Considering the empirical distribution of the survey-estimated variances within each state, one implicitly accounts for the effect of agriculture intensity on the yield. In the case of large states with, for example, varying climatic regions and agriculture intensities, one might want to consider partitioning the state based on the agriculture intensity.

In each approach, the dataset is partitioned into two groups: counties with "good" input data for the model that are indexed by $\Theta^K$ and counties with positive survey estimates and sampling variances beyond a predetermined threshold that are indexed by $\Theta^N$. A set of updated variances is generated for the counties in $\Theta^N$. Then, the updated survey summaries are fed into the subarea-level model in Equation (4) to produce the final modeled county estimates. The results from the two proposed approaches, the modeled estimates produced

based on the original model in Equation (5) and the survey's direct estimates are compared and discussed in Section 4.

*3.1. A Bayesian Approach for Generating Sampling Variances in the Yields for Anomalous Counties*

Let $i = 1, \ldots, n$ be an index for county $i$ in a state. The survey's direct estimate of yield in county $i$ is denoted by $\hat{\theta}_i$, and the sampling variance is denoted by $\hat{\sigma}_i^2$. The auxiliary data used in the models are denoted by $x_i$, including an intercept. Let the thresholds (i.e., lower and upper limits) for the survey estimated variances be $\gamma_1$ and $\gamma_2$ such that

$$
\begin{aligned}
\gamma_1 &= \max\{\min(\hat{\sigma}_i^2), \hat{Q}_{0.25} - 1.5\ IQR(\hat{\sigma}_i^2)\}, \\
\gamma_2 &= \min\{\max(\hat{\sigma}_i^2), \hat{Q}_{0.75} + 1.5\ IQR(\hat{\sigma}_i^2)\},
\end{aligned}
\tag{1}
$$

where $\hat{Q}_{0.25}$ and $\hat{Q}_{0.75}$ are the first and third empirical quartiles of the variances in yield, respectively, computed from the CAPS in a given state. In other words, $\gamma_1 = \max\{\min(\hat{\sigma}_i^2), L\}$ and $\gamma_2 = \min\{\max(\hat{\sigma}_i^2), U\}$, where the boundaries ($L$ and $U$) are the ones defined in Section 2.1.

Then, $n$ counties can be separated into two groups. One group, consisting of counties with variances that exceed the threshold, is indexed by $\Theta^N = \{i : 0 \leq \hat{\sigma}_i^2 \leq \gamma_1,\ \sigma_i^2 \geq \gamma_2\}$, and the other group, consisting of counties with "normal" variances (i.e., within the threshold), is indexed by $\Theta^K = \{i : \gamma_1 < \hat{\sigma}_i^2 < \gamma_2\}$. In the proposed model set-up, we assume for the counties in group $\Theta^K$ that the survey variances are known, and for the counties in group $\Theta^N$, the survey variances are unknown and need to be generated.

Consider the following HB model:

$$
\begin{aligned}
\hat{\theta}_i | \theta_i, \hat{\sigma}_i^2 &\overset{ind}{\sim} N(\theta_i, \hat{\sigma}_i^2),\ i \in \Theta^K, \\
\hat{\theta}_i | \theta_i, \sigma_i^2 &\overset{ind}{\sim} N(\theta_i, \sigma_i^2),\ i \in \Theta^N, \\
\theta_i | \boldsymbol{\beta}, \delta^2 &\overset{ind}{\sim} N(x_i' \boldsymbol{\beta}, \delta^2), i = 1, \ldots, n,
\end{aligned}
\tag{2}
$$

where $(\boldsymbol{\beta}, \sigma^2, \delta^2)$ is a set of nuisance parameters. The mathematical features of Equation (2) are discussed in Appendix A. In addition, the Gibbs sampler is attached in Appendix B.

A diffused prior (i.e., a bivariate normal prior distribution with a fixed and known mean $\hat{\boldsymbol{\beta}}$ and variance and covariance matrix $1000\hat{\Sigma}_{\hat{\beta}}$) is adopted for the coefficients $\boldsymbol{\beta}$:

$$
\boldsymbol{\beta} \sim MN(\hat{\boldsymbol{\beta}}, 1000\hat{\Sigma}_{\hat{\beta}}).
$$

Here, $\hat{\boldsymbol{\beta}}$ are the least squares estimates of $\boldsymbol{\beta}$ obtained from fitting a simple linear regression model of the county-level survey estimates on the auxiliary data $x_i$, and $\hat{\Sigma}_{\hat{\beta}}$ is the estimated covariance matrix of $\hat{\boldsymbol{\beta}}$.

The prior distribution chosen for $\sigma_i^2$ is

$$
\pi(\sigma_i^2) \propto \frac{1}{\sigma_i^2},\ i \in \Theta^N.
$$

Meanwhile, the prior distribution chosen for $\delta^2$ is

$$
\pi(\delta^2) \propto \frac{1}{\delta^2}.
$$

*3.2. A Non-Parametric Approach to Generating Sampling Variances for Anomalous Counties*

Let $i = 1, \ldots, n$ be an index for county $i$ with a positive direct estimate (e.g., in our case study, this is the survey estimated yield) in a given state, and let $\hat{\sigma}_i^2$ represent the sampling variance of the yield for county $i$. The "smoothing" approach based on bootstrap replicates

consists of resampling with replacement $n$ samples from the set of counties with "good" sampling variances; that is, they satisfy the following inequality:

$$\gamma_1 < \hat{\sigma}_i^2 < \gamma_2, \tag{3}$$

where bounds $\gamma_1$ and $\gamma_2$ are defined in Equation (1). These bounds are computed for a given state in our case study based on the variances in the yield obtained from the 2016 CAPS.

The order statistics of the resampled variances for a given state are considered as realizations (of the quantiles) from the empirical distribution of the sampling variances of the yield at the county level. These order statistics, obtained after each resampling iteration, are recorded as columns of a matrix **A** of a size $n \times B$, where $B$ is the number of bootstrap iterations. The averages from each row in **A**, taken over the $B$ bootstrap iterations (i.e., obtained by averaging through the columns of **A**), are

$$\tilde{\sigma}_{(i)}^2 = \frac{1}{B} \sum_{b=1}^{B} a_{i,b},$$

where $i = 1, \ldots, n$ can be considered in lieu of the empirical distribution of the survey variances. These quantities are used to replace their corresponding ordered values of the original sampling variances $\hat{\sigma}_{(i)}^2$ and therefore smooth the sampling variances of the anomalous counties.

## 4. Case Study

### 4.1. Subarea-Level Models of the NASS

The subarea-level models were first developed by Fuller and Goyeneche [18] and later studied in a frequentist framework by Torabi and Rao [19] and Rao and Molina [20]. Erciulescu et al. [5] presented a subarea-level model in a Bayesian framework, where the area represents the agricultural statistics district (groups of neighboring counties within a state, hereafter denoted as ASDs) and the subarea represents the county, and study its performance under different scenarios of data availability.

Let $i = 1, \ldots, m$ be an index for $m$ ASDs in the state, $j = 1, \ldots, n_i^c$ be an index for the $n_i^c$ counties in the $i$th ASD and $n_{ij}$ be the sample size of the $j$th county in the $i$th ASD. The total number of counties within a state is $\sum_{i=1}^{m} n_i^c = n^c$, and the state sample size is $\sum_{i=1}^{m} \sum_{j=1}^{n_i^c} n_{ij} = n$. The direct estimate in county $j$ within the $i$th ASD is denoted by $\hat{\theta}_{ij}$, and the associated variance estimated from the survey is denoted by $\hat{\sigma}_{ij}^2$. Illustrated for one state, one commodity and one parameter (i.e., yield), the model is

$$
\begin{aligned}
\hat{\theta}_{ij}|\theta_{ij}, \hat{\sigma}_{ij}^2 &\overset{ind}{\sim} N(\theta_{ij}, \hat{\sigma}_{ij}^2),\ i = 1, \ldots, m, \\
\theta_{ij}|\boldsymbol{\beta}, \sigma_\mu^2 &\overset{ind}{\sim} N(\boldsymbol{x}_{ij}'\boldsymbol{\beta} + v_i, \sigma_\mu^2),\ j = 1, \ldots, n_i^c, \\
v_i|\sigma_v^2 &\overset{iid}{\sim} N(0, \sigma_v^2),
\end{aligned}
\tag{4}
$$

where $(\boldsymbol{\beta}, \sigma_\mu^2, \sigma_v^2)$ is a set of nuisance parameters.

A diffused prior is adopted for the vector parameter (coefficients) $\boldsymbol{\beta}$ (i.e., a bivariate normal prior distribution with a fixed and known mean and a variance and covariance matrix $1000\hat{\Sigma}_{\hat{\beta}}$), where $\boldsymbol{\beta} \sim MN(\hat{\boldsymbol{\beta}}, 1000\hat{\Sigma}_{\hat{\beta}})$. Here, $\hat{\boldsymbol{\beta}}$ are the least squares estimates of $\boldsymbol{\beta}$ obtained from fitting a simple linear regression model of the county-level direct estimates from the survey on the auxiliary data $\boldsymbol{x}_{ij}$, and $\hat{\Sigma}_{\hat{\beta}}$ is the estimated covariance matrix of $\hat{\boldsymbol{\beta}}$. Identical non-informative prior distributions (i.e., Uniform$(0, 10^{10})$) are adopted for $\sigma_\mu^2$ and $\sigma_v^2$. For more details on the choice of priors for the variance components in the Bayesian models, see the discussion by Browne and Draper [21] and Gelman [22].

NASS publishes the yields to the nearest tenth of a bushel per acre. The variances in yield estimated from the 2016 CAPS for corn were smaller than 0.01 bushels per acre for 215 counties. The nearest neighbor imputation technique is applied to fill in the missing data

pairs, namely the missing direct estimates and missing corresponding sampling variances that will be used to feed the HB model in Equation (4). However, with this approach, counties with non-zero direct estimates and missing or equal-to-zero sampling variances are not modeled. Furthermore, the model in Equation (4) cannot produce estimates (or if produced, the estimates are unreliable) for the 348 counties with valid (positive) direct survey estimates and corresponding sampling variances outside the threshold bounds ($L$ and $U$).

In more recent research on modeling yields at the county level in the NASS, counties with valid direct estimates that are below some threshold sampling variances are considered in the model. Instead of excluding them from the subarea-level model in Equation (4), these unreliable sampling variances are assumed to be unknown (the same applies throughout all anomalous counties) and are updated using a Bayesian technique. This approach is formally presented by the updated subarea-level model in Equation (5). In what follows, we briefly describe this research.

Let $n$ counties be separated into two groups. One group, consisting of counties with known direct estimates and sampling variances below some threshold (e.g., less than 1 bushel per acre for corn), is indexed by $\Theta^M = \left\{ i, j : \hat{\sigma}_{ij}^2 \leq 1 \right\}$. The other group, consisting of counties with known direct estimates and sampling variances above the threshold, is indexed by $\Theta^A = \left\{ i, j : \hat{\sigma}_{ij}^2 > 1 \right\}$.

Then, the updated subarea-level model (hereafter addressed as the original model) is

$$
\begin{aligned}
\hat{\theta}_{ij} | \theta_{ij}, \hat{\sigma}_{ij}^2 &\stackrel{ind}{\sim} N(\theta_{ij}, \hat{\sigma}_{ij}^2), \ i, j \in \Theta^A, \\
\hat{\theta}_{ij} | \theta_{ij}, \delta^2 &\stackrel{ind}{\sim} N(\theta_{ij}, \delta^2), \ i, j \in \Theta^M, \\
\theta_{ij} | \boldsymbol{\beta}, v_i, \sigma_\mu^2 &\stackrel{ind}{\sim} N(\boldsymbol{x}_{ij}' \boldsymbol{\beta} + v_i, \sigma_\mu^2), \ j = 1, \ldots, n_i^c, \\
v_i | \sigma_v^2 &\stackrel{iid}{\sim} N(0, \sigma_v^2), i = 1, \ldots, m,
\end{aligned}
\tag{5}
$$

where $(\boldsymbol{\beta}, \delta^2, \sigma_\mu^2, \sigma_v^2)$ is a set of nuisance parameters.

The priors adopted for $(\boldsymbol{\beta}, \sigma_\mu^2, \sigma_v^2)$ are the same as those in Equation (4), and the prior adopted for $\delta^2$, the unknown constant variance throughout all anomalous counties, is $\pi(\delta^2) \propto \frac{1}{\delta^2}$.

The limitations of the original model in Equation (5) are twofold. First, the model assumes that all unknown variances are the same, and second, not all unreliable sampling variances are considered by the model. The variances with values that fall in the upper extreme right tail of the distribution of sampling variances are unreliable and not considered by Equation (5).

This paper addresses the challenge of improving the unreliable sampling variances for counties with valid (positive) direct survey estimates more realistically. By considering all unreliable sampling variances in both tails of the distribution (i.e., outside of the threshold bounds $L$ and $U$) and relaxing the assumption of constant (unreliable) variances throughout small areas, this research overcomes the limitations of the original model shown in Equation (5). One can apply any of the two alternative approaches presented earlier in Section 3 to improve the HB model inputs. As an illustration, our case study shows that more reliable final estimates of the yield were produced for the US counties by the HB model fed with updated survey summaries (based on the two alternative approaches presented in this paper) when compared with the existing approaches.

*4.2. Results*

In this section, nationwide results from different estimation procedures that used CAPS data to produce the county-level corn yield estimates for 2016 are presented. We compare the corn yield estimates and the associated measures of uncertainty produced from the following:

- A survey;
- The original model in Equation (5);
- The updated model in Equation (4) using improved sampling variances based on the Bayesian method as the input;
- The updated model in Equation (4) using improved sampling variances based on the bootstrap method as the input.

The Markov chain Monte Carlo (MCMC) simulation method was used to fit the Bayesian models using R and JAGS [23]. The JAGS model descriptions used in the R script are shown in Appendix C. All the Bayesian models are fit for each state individually, and there were 37 corn states in the 2016 CAPS. In each model, three chains were run for our MCMC simulation. Each chain contained 10,000 Monte Carlo samples, and the first 2000 iterates were discarded as a burn-in to improve the mixing of each chain. In order to eliminate the correlations among the neighboring iterations, those iterations were thinned by taking a systematic sample of one in every eight samples. Finally, 1000 MCMC samples in each chain were obtained for constructing the posterior distributions of the parameters and make inferences for the yield estimates. Convergence diagnostics were conducted to make sure that the MCMC samples were mixing well. The convergence was monitored using trace plots, the multiple potential scale reduction factors ($\hat{R}$ close to one) and the Geweke test of stationarity for each chain (see [24,25]). We found that the Geweke tests for all the parameters in models (2), (4) and (5) were not significant, and the effective sample sizes were all near the actual sample size of 1000. (Nearly all of them were 1000.)

For the bootstrap method, $B = 1000$ samples of a size $n$ were used to construct the empirical distribution of the sampling variances (see Appendix D). Then, as described in Section 3.2, the new set of values was drawn from the empirical distribution to update the unreliable variances for the "anomalous" counties within each state. The updated sampling variances obtained via the bootstrap method satisfied the inequalities in model (3) and provided more reasonable values for the extreme sampling variances, which were further used as inputs in the model in Equation (4).

We recall here that the 2016 CAPS sample consists of 37 states comprising 2329 counties with positive yields or production for corn. There were 99 counties with zero sampling variances, 142 counties with positive sampling variances below $L$ and 107 counties with sampling variances that were relatively large, being greater than $U$. In total, there were 348 anomalous counties with sampling variances falling outside the bounds $L$ and $U$. These bounds defined, earlier in Section 2.1, vary by state.

Before comparing the final modeled estimates of the yields generated from all methods discussed in this paper, we briefly show the improvement gained for the sampling variances by applying the two proposed methods to the 348 anomalous counties. Table 1 shows the five-number summary (i.e., minimum, first, second and third quartile, as well as the maximum) of the survey-estimated variances, improved sampling variances based on the Bayesian method and improved sampling variances based on the bootstrap approach at the county level for the anomalous counties. It is straightforward to see that the survey-estimated variances were highly right-skewed. A large part of those was very close to zero or extremely small. The first quartile was 0.00, the median was $6.25 \times 10^{-14}$, the third quartile was 0.71, and the maximum was 5264.01. However, the variances generated from both the Bayesian and bootstrap methods were improved. These updated variances appeared to not be that extreme (i.e., they were far from zero), shifted more to the right and were more centered than the survey-estimated variances. The minimum and first quartile of the updated variances based on the bootstrap method were smaller than ones generated by the Bayesian method (Table 1). The median, third quartile and maximum of the updated variances based on the Bayesian method were smaller than the ones from the bootstrap method.

**Table 1.** Five-number summary of the sampling variances, improved sampling variances based on the Bayesian method and improved sampling variances based on the bootstrap approach at the county level in all anomalous counties.

| | Statistics | Survey | Bayesian | Bootstrap |
|---|---|---|---|---|
| Anomalous Counties | Min | 0.00 | 10.09 | 1.74 |
| | 1st Qu. | 0.00 | 83.13 | 60.33 |
| | Median | $6.25 \times 10^{-14}$ | 124.05 | 142.36 |
| | 3rd Qu. | 0.71 | 167.16 | 289.66 |
| | Max | 5264.01 | 1426.81 | 1685.06 |

The performance of each approach was evaluated based on the relative bias of the final modeled estimates produced from each method toward the published estimates. The absolute relative differences (ARDs) between the estimates from any procedure and the published estimates were computed as follows:

$$\text{ARD} = 100 \times \left| \frac{\tilde{\theta} - \theta^P}{\theta^P} \right|,$$

where $\tilde{\theta}$ is the final modeled estimate or the survey's direct estimate and $\theta^P$ is the corresponding published county-level yield estimate.

Table 2 shows the nationwide results using a five-number summary of the ARDs of the yield estimates produced from all four approaches, with a focus on the published estimates for the anomalous counties and all available counties afterward. The median of the ARDs when the Bayesian and bootstrap methods qwre applied in the anomalous counties were 7.93% and 6.71%, respectively. These were much smaller than the median ARD of the survey's direct estimates and, to a lesser extent, smaller than the median ARD of the estimates produced from the original model in Equation (5). The estimates based on the Bayesian and bootstrap methods were generally closer to the published estimates in the anomalous counties. Similar relationships can be seen in the third quartile. However, the maximum ARD of the survey's direct estimates was the smallest of all the other methods, and this was because some of the direct estimates from the survey were missing, and other methods provided a complete dataset. The median ARDs from the bootstrap approach were the smallest in the anomalous counties. The maximum ARD of the estimates produced by the Bayesian method was the smallest in the anomalous counties. In all counties, all modeled estimates were generally closer to the published estimates when compared with the direct survey estimates. Overall, Table 2 reveals an improvement in performance for the HB models in small areas under the two proposed approaches. All five-number summaries of the ARDs of the yield estimates based on the Bayesian and bootstrap methods were smaller than those from the original model.

In addition, the choropleth maps (Figure 3) depict the ARDs for the county-level estimates produced from different methods in selected states, known as the corn belt states for dominating the corn production in the US. As the difference between the estimates produced from each method and the published estimates increased (relative to the published estimate), the corresponding colored area became darker. Most counties are shown as yellow, indicating that the estimates produced by the model were closer to the published estimates. Counties shown from dark green to blue or purple on the map depicting the estimates based on the survey (upper left corner) consisted of very small sample sizes and unreliable sampling variances for the yield. The corresponding counties in other maps, which depict the estimates based on the subarea-level models (original model in Equation (5) and the model in Equation (4) with updated inputs) appeared to be much lighter. For the areas with small sample sizes, the subarea-level models produced the yield estimates by incorporating other (administrative) data and by "borrowing information" across and within areas and subareas.

**Table 2.** Five-number summary of the absolute relative differences (%) of the published estimates and the estimates from the survey, original model and updated model using the improved sampling variances based on the Bayesian method and bootstrap method as input, computed at the county level in the US.

|  | Statistics | Survey | Original | Bayesian | Bootstrap |
|---|---|---|---|---|---|
| Anomalous Counties | Min | 0.00 | 0.02 | 0.02 | 0.01 |
|  | 1st Qu. | 3.63 | 4.96 | 2.62 | 2.43 |
|  | Median | 21.05 | 11.65 | 7.93 | 6.71 |
|  | 3rd Qu. | 100.00 | 24.00 | 21.95 | 22.60 |
|  | Max | 140.66 | 199.70 | 187.43 | 195.60 |
| All Counties | Min | 0.00 | 0.00 | 0.00 | 0.00 |
|  | 1st Qu. | 0.60 | 0.60 | 0.57 | 0.51 |
|  | Median | 1.85 | 1.73 | 1.62 | 1.48 |
|  | 3rd Qu. | 5.45 | 5.13 | 4.46 | 4.06 |
|  | Max | 140.66 | 199.70 | 187.43 | 195.60 |

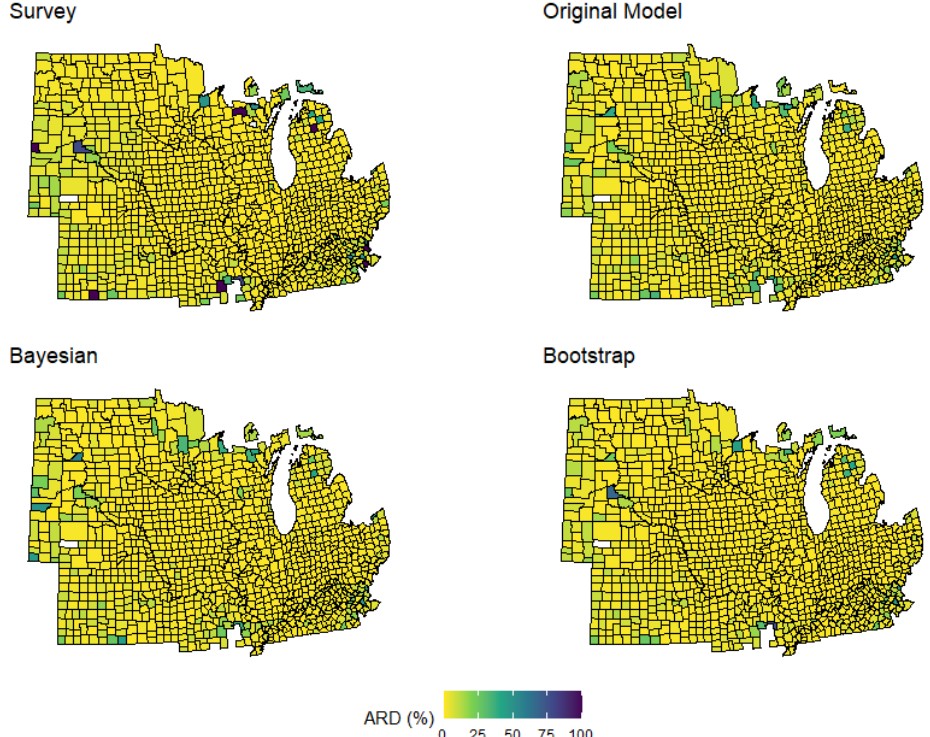

**Figure 3.** Absolute relative differences (%) of the published estimates and the estimates from the survey, original model and updated model using the improved sampling variances based on the Bayesian method and bootstrap method as the input, computed at the county level for select states.

The correlation matrix of the published estimates of the yield, survey's direct estimates of the yield, estimates of the yield based on the original model in Equation (5), estimates of the yield from the model in Equation (4) with improved sampling variances based on the Bayesian method as the input and estimates of the yield from the model in Equation (4) with improved sampling variances based on the bootstrap method as the input are shown in Table 3. All the correlations were larger than 0.75, indicating high correlation among the final estimates from all methods. The highest correlation with the published yield estimates appeared for the estimates produced from the original model in Equation (5). This was expected, since the published estimates were produced using several sources of information where the original model (currently in production) plays a central role. Furthermore, the correlation between the survey and the published estimates was the

lowest, since the direct survey estimates did not leverage model-based solutions designed to improve the estimation accuracy. Table 3 also indicates that all model-based estimates were more accurate than the survey estimates.

**Table 3.** Correlations among the published estimates of the yield, survey's direct estimates of the yield, estimates of the yield based on the original model, estimates of the yield based on the Bayesian method and estimates of the yield based on the bootstrap method at the county level in the US.

|  | Survey | Original Method | Bayesian Method | Bootstrap Method |
|---|---|---|---|---|
| Published | 0.8627 | 0.9577 | 0.9566 | 0.9485 |
| Survey |  | 0.8611 | 0.8716 | 0.8894 |
| Original Method |  |  | 0.9897 | 0.9790 |
| Bayesian Method |  |  |  | 0.9913 |

Table 4 shows the five-number summaries of the coefficients of variation (CVs) of the county-level yield estimates from our case study, produced nationwide using the four approaches discussed in this paper. We recall that there were 99 counties in the 2016 CAPS with zero sampling variances for the yield. Hence, the CVs from the survey for these counties were not valid statistics, and these counties were removed from the CV comparison. The CVs of the yield estimates from the survey among the anomalous counties consisted of extreme values close to either zero or one. However, the CVs of the yield estimates from the original model and Bayesian and bootstrap methods were more stable than the CVs from the survey. The bootstrap method provided the smallest CVs among all three methods. All the five-number summaries from the original model were larger than the two alternative methods proposed in this paper. Over all counties, one can observe the decrease in CVs (an increase in relative precision) from the models based on three approaches when compared with the survey CVs. The original model had the smallest first quartile CVs (2.51%). The smallest median, third quartile and the smallest maximum CVs were shown when the bootstrap method was used. The results demonstrate the tendency of the small area models to improve the accuracy of the estimates when compared with the accuracy of the survey estimates, especially in areas with small sample sizes (i.e., counties with very large CVs).

**Table 4.** Five-number summary of coefficients of variation (CVs, %) of the yield estimates from the survey, original model and the updated model using the improved sampling variances based on the Bayesian method and the bootstrap method as input, computed at the county level in the US.

|  | Statistics | Survey | Original | Bayesian | Bootstrap |
|---|---|---|---|---|---|
| Anomalous Counties | Min | $1.01 \times 10^{-7}$ | 9.29 | 6.15 | 4.23 |
|  | 1st Qu. | $5.00 \times 10^{-7}$ | 14.13 | 8.85 | 8.27 |
|  | Median | 0.25 | 16.36 | 10.85 | 9.88 |
|  | 3rd Qu. | 61.25 | 21.07 | 13.63 | 10.47 |
|  | Max | 97.70 | 46.42 | 39.36 | 16.06 |
| All Counties | Min | $1.01 \times 10^{-7}$ | 0.70 | 0.70 | 0.76 |
|  | 1st Qu. | 2.59 | 2.51 | 2.54 | 2.54 |
|  | Median | 4.41 | 4.26 | 4.30 | 4.20 |
|  | 3rd Qu. | 8.27 | 7.56 | 7.71 | 7.31 |
|  | Max | 97.70 | 46.42 | 52.93 | 40.09 |

Note: Ninety-nine counties with zero survey CVs were removed from this table. There were 249 anomalous counties.

## 5. Discussion and Final Remarks

This paper introduces two alternative approaches to mitigating unreliable sampling variances. In the parametric Bayesian method, we adopt non-informative priors to model the unreliable sampling variances. The posterior means of the variance estimates are used as inputs in the subarea-level model to generate the final modeled estimates of interest.

In the non-parametric bootstrap approach, we construct the empirical distribution of the sampling variances by resampling without replacement from the reliable variances. Drawings from this distribution are used to smooth the unreliable sampling variances, which were further used as input for the HB subarea model to generate the final modeled estimates of interest. NASS's 2016 CAPS yield data for corn are used to illustrate each approach. The final modeled estimates for the year 2016 at the county level are computed using the updated sampling variances, which are produced based on each approach.

The novelty of the work presented in this article stands on introducing a strategy to improve the performance of the HB small area models (which take survey summaries and other auxiliary data as input) wherever the relationship between the survey-estimated variances and direct estimates cannot be modeled using classical approaches. Two (statistically sound) alternatives, one parametric (Bayesian modeling under the area-level model set-up) and the other non-parametric (distribution-free technique using bootstrap sampling) for mitigating the unreliable (beyond some threshold) survey-estimated variances are explored.

For ease of understanding and illustration, the two methods are applied to data from the USDA's NASS crops county estimates program (presented in Section 4). The original model in Equation (5) treats all sampling variances smaller than one as unknown and updates these variances into reliable inputs for the HB small area models. However, the original model does not consider the extremely large sampling variances that are experienced in small areas. The two proposed approaches described in Section 3 take all unreliable sampling variances into consideration and smooth these variances into reliable inputs for the HB subarea model. The results from the case study in Section 4 show the improvement in the sampling variances among the anomalous counties. In addition, substantial improvement is shown in the final modeled estimates produced based on the updated sampling variances, as well as an increase of approximately 10% in the number of final model-based county estimates produced. This has the potential to increase the number of counties published by the NASS as official statistics and, most importantly, with the associated measures of uncertainty.

The techniques presented in this paper for improving the modeled estimates for small domains are not limited to the application of HB small area models to agricultural data. Both algorithms are general in nature (as described in Section 3) and can be applied to any survey that suffers from small area issues. Applying these techniques assures the relevance of using the HB models in small areas where estimates from surveys are missing or not reliable and cannot be modeled using classical approaches. Improving the sampling variances would allow for the HB small area models to achieve their intended objective of producing reliable (and reproducible) estimates. In sum, we show that the issue with HB models failing to provide reliable model-based estimates for small areas is solvable, even in scenarios where the relationship between the survey-estimated variances and direct estimates cannot be modeled using existing classical approaches. This paper provides a strategy for the solution.

**Author Contributions:** Conceptualization, V.B. and B.N.; methodology, B.N., V.B., L.C. and L.S.; software, L.C., L.S. and H.B.; validation, L.C.; formal analysis, L.C.; investigation, V.B. and L.C.; writing—original draft preparation, V.B., L.C., L.S. and H.B.; writing—review and editing, V.B., L.C., L.S., H.B. and B.N.; visualizations, L.C. and L.S.; supervision, V.B. and B.N. All authors have read and agreed to the published version of the manuscript.

**Funding:** This research was supported by the U.S. Department of Agriculture's National Agricultural Statistics Service. Balgobin Nandram was supported by a grant from the Simons Foundation (#353953, Balgobin Nandram).

**Institutional Review Board Statement:** Not applicable.

**Informed Consent Statement:** Not applicable.

**Data Availability Statement:** Not applicable.

**Acknowledgments:** The findings and conclusions in this paper are those of the authors and should not be construed to represent any official USDA or US government determination or policy.

**Conflicts of Interest:** The authors declare no conflict of interest.

## Appendix A. Mathematical Feasures for Model (2)

It is pertinent to give a mathematical explanation of the features of the model in Equation (2). For this discussion, we assume that $\boldsymbol{\beta}$ and $\delta^2$ are fixed but unknown. We observe that Equation (2) has two submodels. For $i \in \Theta^K$, we have

$$\hat{\theta}_i \mid \theta_i, \hat{\sigma}_i^2 \overset{ind}{\sim} N(\theta_i, \hat{\sigma}_i^2),$$

$$\theta_i \mid \boldsymbol{\beta}, \delta^2 \overset{ind}{\sim} N(\boldsymbol{x}_i'\boldsymbol{\beta}, \delta^2),$$

For $i \in \Theta^N$, we have

$$\hat{\theta}_i \mid \theta_i, \sigma_i^2 \overset{ind}{\sim} N(\theta_i, \sigma_i^2),$$

$$\theta_i \mid \boldsymbol{\beta}, \delta^2 \overset{ind}{\sim} N(\boldsymbol{x}_i'\boldsymbol{\beta}, \delta^2).$$

The conditional posterior distributions of $\theta_i$ are

$$\theta_i \mid \hat{\theta}_i, \hat{\sigma}_i^2 \overset{ind}{\sim} N\{\lambda_i \hat{\theta}_i + (1-\lambda_i)\boldsymbol{x}_i'\boldsymbol{\beta}, (1-\lambda_i)\delta^2\}, \lambda_i = \frac{\delta^2}{\delta^2 + \hat{\sigma}_i^2}, i \in \Theta^K,$$

and

$$\theta_i \mid \hat{\theta}_i, \sigma_i^2 \overset{ind}{\sim} N\{\lambda_i \hat{\theta}_i + (1-\lambda_i)\boldsymbol{x}_i'\boldsymbol{\beta}, (1-\lambda_i)\delta^2\}, \lambda_i = \frac{\delta^2}{\delta^2 + \sigma_i^2}, i \in \Theta^N.$$

The main difference in these two distributions occurs in $\lambda_i$ (i.e., $\hat{\sigma}_i^2$ versus $\sigma_i^2$).

In most applications, the cardinality of $\Theta^K$ is much larger than that of $\Theta^N$; that is, the data from $\Theta^K$ will dominate those from $\Theta^N$, and therefore, we can approximate $\boldsymbol{\beta}$ and $\delta^2$ using the data from only $\Theta^K$. For example, denote the maximum likelihood estimators (MLEs) as $\hat{\boldsymbol{\beta}}$ and $\hat{\delta}^2$. Then, the MLEs from $\Theta^K$ are substituted into $\Theta^N$ to obtain the adjusted model:

$$\hat{\theta}_i \mid \theta_i, \sigma_i^2 \overset{ind}{\sim} N(\theta_i, \sigma_i^2),$$

$$\theta_i \overset{ind}{\sim} N(\boldsymbol{x}_i'\hat{\boldsymbol{\beta}}, \hat{\delta}^2),$$

where we assume that $\hat{\boldsymbol{\beta}}$ and $\hat{\delta}^2$ are known. It is now easy to show that

$$\hat{\theta}_i \mid \sigma_i^2 \overset{ind}{\sim} N(\boldsymbol{x}_i'\hat{\boldsymbol{\beta}}, \hat{\delta}^2 + \sigma_i^2), i \in \Theta^N.$$

Finally, by reparameterizing this latter density, we can obtain the MLEs of the $\sigma_i^2$ from

$$\pi(\hat{\theta}_i \mid \sigma_i^2) = \sqrt{\frac{1}{2\pi(\hat{\delta}^2 + \sigma_i^2)}} \exp\left\{-\frac{1}{2(\hat{\delta}^2 + \sigma_i^2)}(\hat{\theta}_i - \boldsymbol{x}_i'\boldsymbol{\beta})^2\right\}$$

$$= \sqrt{\frac{\phi_i}{2\pi\hat{\delta}^2}} \exp\left\{-\frac{\phi_i}{2\hat{\delta}^2}(\hat{\theta}_i - \boldsymbol{x}_i'\boldsymbol{\beta})^2\right\}, i \in \Theta^N,$$

where $\phi_i = \frac{\hat{\delta}^2}{(\hat{\delta}^2 + \sigma_i^2)}, i \in \Theta^N$ and $0 < \phi_i < 1$. Now, the MLEs of $\phi_i$ can be obtained, which we denote by $\hat{\phi}_i$, and so we find the MLEs of $\sigma_i^2$ as follows:

$$\hat{\sigma}_i^2 = \frac{1 - \hat{\phi}_i}{\hat{\phi}_i}\hat{\delta}^2.$$

Note that $\hat{\phi}_i^* = \hat{\delta}^2/(\hat{\theta}_i - x_i\beta)^2$ are not really MLEs because the numerator may be zero and $\hat{\phi}_i^*$ may not be in the open interval $(0, 1)$. Therefore, when letting $\hat{\lambda}_i = \hat{\delta}^2/(\hat{\delta}^2 + \hat{\sigma}_i^2)$, we have

$$\theta_i \mid \hat{\theta}_i, \hat{\beta}, \hat{\delta}^2, \hat{\sigma}_i^2 \overset{ind}{\sim} N\{\hat{\lambda}_i x_i'\hat{\beta} + (1 - \hat{\lambda}_i)\hat{\theta}_i, (1 - \hat{\lambda}_i)\hat{\delta}^2\}, i \in \Theta^N.$$

Now, an inference can be made about the $\theta_i, i \in \Theta^N$. The Gibbs sampler, a more coherent procedure for fitting Equation (2), is given in Appendix B.

## Appendix B. Gibbs Sampler for Equation (2)

Here, we show the Gibbs sampler for Equation (2) when $i \in \Theta^K$:

$$\hat{\theta}_i \mid \theta_i, \hat{\sigma}_i^2 \overset{ind}{\sim} N(\theta_i, \hat{\sigma}_i^2),$$

$$\theta_i \mid \beta, \delta^2 \overset{ind}{\sim} N(x_i'\beta, \delta^2),$$

$$\beta \sim MN(\hat{\beta}, 1000\hat{\Sigma}_{\hat{\beta}}),$$

$$\pi(\delta^2) \propto \frac{1}{\delta^2}.$$

The joint posterior density of $\theta_i, \beta, \delta^2$ is given by

$$\pi(\theta_i, \beta, \delta^2 | \hat{\theta}_i, \hat{\sigma}_i^2) \propto \frac{1}{\delta^2} \prod_{i \in \Theta^K} \left[ \frac{1}{\sqrt{\hat{\sigma}_i^2}} \exp\left\{ -\frac{1}{2\hat{\sigma}^2}(\hat{\theta}_i - \theta_i)^2 \right\} \times \frac{1}{\sqrt{\delta^2}} \exp\left\{ -\frac{1}{2\delta^2}(\theta_i - x_i'\beta)^2 \right\} \right]$$

$$\times \sqrt{\det(\hat{\Sigma}_{\hat{\beta}})} \times \exp\left\{ -\frac{1}{2}(\beta - \hat{\beta})'(1000\hat{\Sigma}_{\hat{\beta}})^{-1}(\beta - \hat{\beta}) \right\}.$$

The full conditional distributions for Gibbs sampling are as follows:

1.  $\beta | \theta, \delta^2 \sim MVN\left( \Sigma_\beta \left( \frac{\sum_{i \in \Theta^K} \theta_i x_i}{\delta^2} + (1000\hat{\Sigma}_{\hat{\beta}})^{-1}\hat{\beta} \right), \Sigma_\beta \right)$,

    where $\Sigma_\beta = \left( \frac{\sum_{i \in \Theta^K} x_i x_i'}{\delta^2} + (1000\hat{\Sigma}_{\hat{\beta}})^{-1} \right)^{-1}$;

2.  $\theta_i | \beta, \delta^2 \overset{ind}{\sim} N(\lambda_i \hat{\theta}_i + (1 - \lambda_i)x_i'\beta, (1 - \lambda_i)\delta^2)$, $\lambda_i = \frac{\delta^2}{\delta^2 + \hat{\sigma}_i^2}$;

3.  $\delta^2 | \theta, \beta \sim IG\left( \frac{n^*-1}{2}, \frac{1}{2}\sum_{i \in \Theta^K}(\theta_i - x_i'\beta)^2 \right)$, where $n^*$ is the size of $\Theta^K$.

## Appendix C. RJAGS Codes for the Models in Equations (4) and (5)

The following is the JAGS model description of the subarea model in Equation (4) which was used in the R script.

```
model{
  Xbeta <- cX%*%beta
  for(j in 1:n){
    #county
      thetahatij[j] ~ dnorm(thetaij[j], 1/vhat.dirij[j])
      thetaij[j] ~ dnorm(thetaij0[j], sigma2u.inv)
      thetaij0[j] <- Xbeta[j] + vi[id[j]]
  }
  ## Priors:
  for (i in 1:m){
      vi[i] ~ dnorm(0,sigma2v.inv)
  }
  sigma2v ~ dunif(0, 10^10)
  sigma2v.inv <- pow(sigma2v, -1)
  sigma2u ~ dunif(0, 10^10)
  sigma2u.inv <- pow(sigma2u, -1)

  beta ~ dmnorm(betahat,Sigmahatbeta.inv)
  Sigmahatbeta.inv <- inverse(Sigmahatbeta*10^3)
}
```

The following is the JAGS model description of the subarea model in Equation (5) which was used in the R script.

```
model{
  Xbeta <- cX%*%beta
  for(j in 1:nt){
    #county
    thetahatij[nc[j]] ~ dnorm(thetaij[nc[j]],1/vhat.dirij[nc[j
        ]])
    thetaij[nc[j]] ~ dnorm(thetaij0[nc[j]],sigma2u.inv)
    thetaij0[nc[j]] <- Xbeta[nc[j]] + vi[id[nc[j]]]
  }

  for(k in 1:ntm){
    thetahatij[nm[k]] ~ dnorm(thetaij[nm[k]], tau)
    thetaij[nm[k]] ~ dnorm(thetaij0[nm[k]], sigma2u.inv)
    thetaij0[nm[k]] <- Xbeta[nm[k]] + vi[id[nm[k]]]
  }
  ## Priors:
  for (i in 1:m){
   vi[i] ~ dnorm(0,sigma2v.inv)
  }

  sigma2v ~ dunif(0, 10^10)
  sigma2v.inv <- pow(sigma2v, -1)

  sigma2u ~ dunif(0, 10^10)
  sigma2u.inv <- pow(sigma2u, -1)

  beta ~ dmnorm(betahat,Sigmahatbeta.inv)
  Sigmahatbeta.inv <- inverse(Sigmahatbeta*10^3)

  tau ~ dgamma(0.001,0.001)
  vhat <- pow(tau,-1)
}
```

## Appendix D. R Code for the Bootstrap Methodology

The following is the code to smooth the sampling variance estimates using the bootstrap approach as implemented in the R script.

```
getBSEstim <- function(ss, sfip, scale = 2, BS_ITER = 1000L) {
  tapply(ss, sfip, function(xx) {
    or <- order(xx)
    nn <- length(xx)
    fn <- fivenum(xx)
    iq <- IQR(xx)
    cc <- c(max(fn[1], fn[2] - scale * iq),
            min(fn[5], fn[4] + scale * iq))
    smp <- xx[xx > cc[1L] & xx < cc[2L]]
    estim <- replicate(BS_ITER, {
      sort(sample(smp, size = nn, replace = TRUE))
    })
    estim <- as.vector(rowMeans(estim))
    estim[or] <- sort(estim)
    return(list(bounds = cc, sample = xx, estim = estim))
  })
}
sigmas <- as.numeric(dta$se2_G)
thetas <- as.numeric(dta$ratio2_G)
res <- getBSEstim(sigmas[thetas > 0],
                  dta$state_cd[thetas > 0], 1.5)
dta$BS_estim <- 0 * sigmas
ina <- is.na(sigmas)
for (i in names(res)) {
  wh <- dta$state_cd == i & thetas > 0
  dta[!ina & wh, ''BS_estim''] <- res[[i]]$estim
}
```

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
