# Peer review of "Smoothing County-Level Sampling Variances to Improve Small Area Models’ Outputs"

_stats, doi:10.3390/stats5030052_

Round 1

Reviewer 1 Report

This paper address a very specific problem related to estimation of local outputs in agriculature in USA. The methods appear to be correct and represent an improvement of the methods presently used by United States Department of Agriculture. The article may be highly relevant for statisticians emplyed by the United States Department of Agriculture, but the article may be of little interest for anybody else. The novelty presented in the article limited to the applied level and represents little or no theoretical progress that may be of more general interest. 

The exposition could be improved by restructuring the material in parts that may be of more general interest and parts that are very only relevant for people working with the exact same type of data.

Reviewer 2 Report

I thank MDPI to offer me a chance to review this paper. 

This paper explores two alternative methods, Bayesian method and bootstrap approach, for mitigating unreliable survey estimated variances of yield for small area domains at the county level. This paper is interesting where an innovative applications of statistics is provided in the disciplines. However, it does need substantial revision before it can be published. 

Although authors claims that the  sampling variances can also be imprecise and unreliable,

the authors did not show clearly using their empirical data, it would be clear if in their introduction section the authors can show the reader this problem  and later solve this problem via their proposed method and results. I also find the paper might be too lengthy. Authors may consider to reorganize and provide a version more concisely describe the background, the problem, the method and the solution. Authors may consider to shorten the introduction section, too.

+ As this is a statistics journal, could authors please provide the trace plot that show the convergence of the parameter estimation as well as the posterior, prior, likelihood plots for their Bayesian approach ?

+ Figures do require more elaboration and enhancement. The current versions may not well meet the requirements.

+ Could author please provide an algorithmic table for describing the statistical procedures?

+ The conclusion is too lengthy, authors may consider to summarize the study via providing several key points. 

+ By MDPI policy, could authors please provide scripts/code for reproducible results (Figure and Table) ?

Reviewer 3 Report

Thanks for the opportunity to review this manuscript. I had an interesting reading of the manuscript which proposed two alternative approaches: Bayesian method and bootstrap approach, for mitigating unreliable survey estimated variances of yield for small area domains at the county level. These approaches were illustrated by the NASS’s 2016 CAPS yield data for corn, and the results show that both the proposed approaches have higher precision. The manuscript is well done. The mathematical equations in this manuscript are clearly written, and the conclusions are justified. These results will be helpful in further small area models. The authors should be commended for their great effort.

 One suggestion is that the tense in the paper should be consistent. For example, in Page 1, “This paper explores different alternatives to mitigate the unreliable (beyond some threshold) sampling variances of yield at the county level.” uses the present tense. However, in page 12, “This paper introduced two alternative approaches to mitigating unreliable sampling variances that were further used as input into HB small area models”, applies the past tense.

Round 2

Reviewer 2 Report

I thank authors to take my comments and make relevant revision. In this version, authors revise their paper, the script and relevant material (flow chart, revised paragraph) are provided. 

However, my worry is that the authors may not clearly show to the audience that  the two different approaches do actually help to solve the problems of unreliable sampling variances. Below are my comments, I hope authors can find them useful.

point: If this research work is  derived from the empirical data, the authors may consider to demonstrate it using the empirical data in the introduction section and then states with evidence how the unreliable sampling variances occurred with a summary statistics.  Then show how their models/methods do mitigate this situation. This organization would be clear and directly raise attention to the readers.

point: L339 vs. Fig 4:  while the survey CV are zeros (not-valid), the CVs produced via the other three approach are nonzero (actually widely spread).  If authors would like to make comparison, authors shall consider to use valid statistics to support their finding (smaller CV from alternative approach). It would be good to remove those zero survey CVs from the data first, and conduct the analysis, and then make the comparison. 
